# Renewable Schiff-Base Ionic Liquids for Lignocellulosic Biomass Pretreatment

**DOI:** 10.3390/molecules27196278

**Published:** 2022-09-23

**Authors:** Hemant Choudhary, Venkataramana R. Pidatala, Mood Mohan, Blake A. Simmons, John M. Gladden, Seema Singh

**Affiliations:** 1Deconstruction Division, Joint BioEnergy Institute, 5885 Hollis Street, Emeryville, CA 94608, USA; 2Department of Bioresource and Environmental Security, Sandia National Laboratories, 7011 East Avenue, Livermore, CA 94551, USA; 3Biological Systems and Engineering Division, Lawrence Berkeley National Laboratory, 1 Cyclotron Road, Berkeley, CA 94720, USA; 4Department of Biomaterials and Biomanufacturing, Sandia National Laboratories, 7011 East Avenue, Livermore, CA 94551, USA

**Keywords:** ionic liquids, vanillin, ethylene diamine, lignocellulose, biofuel, biorefinery, lignin

## Abstract

Growing interest in sustainable sources of chemicals and energy from renewable and reliable sources has stimulated the design and synthesis of renewable Schiff-base (iminium) ionic liquids (ILs) to replace fossil-derived ILs. In this study, we report on the synthesis of three unique iminium-acetate ILs from lignin-derived aldehyde for a sustainable “future” lignocellulosic biorefinery. The synthesized ILs contained only imines or imines along with amines in their structure; the ILs with only imines group exhibited better pretreatment efficacy, achieving >89% sugar release. Various analytical and computational tools were employed to understand the pretreatment efficacy of these ILs. This is the first study to demonstrate the ease of synthesis of these renewable ILs, and therefore, opens the door for a new class of “Schiff-base ILs” for further investigation that could also be designed to be task specific.

## 1. Introduction

Depleting fossil-derived organic solvents are still widely used in daily and industrial activities, especially synthetic chemistry, despite the extensive literature that has described related environmental, health, and safety issues [1]. In this regard, ionic liquids (ILs) (organic cation(s) containing salts with a melting point below 100 °C) have been identified as a promising alternative to organic solvents [2]. Given the ease of tunability, ILs can be designed to accommodate several advantages including chemical and thermal stability, dissolution ability, negligible vapor pressure, among others [3,4,5]. Due to this, ILs have found applications in several areas, for example, catalysis [6,7], biologically actives [8,9], process development [10], energy storage [5,11], energy dense materials [12], biomedical [13], lubricants [14], and others [15]. Although not all reported ILs are renewable, cost-effective, biodegradable, or nontoxic, the potential of limitless combinations of cation(s) and anion(s) facilitate the design and production of distinct ILs with unique physicochemical and desired properties to meet specific applications.

Among the several examples mentioned above, ILs have been found to be exceptional in biopolymer dissolution that facilitates their bioconversion into biofuels [16,17,18,19,20,21]. The application of IL technologies for sustainable processing of biomass to meet the large quantities of biofuels, demands a reliable and renewable source for IL productions. Unfortunately, the majority of ILs produced today rely on rapidly depleting fossil resources, restricting the exploration of the huge potential offered by billions of tons of unused available lignocellulosic biomass feedstock that includes agricultural, forest, and herbaceous residues. The design of new renewable solvents, particularly ILs, from renewable sources, thereby, remains to be an open quest for researchers in the field.

Lignin is an underutilized source of aromatics that is largely utilized for generating heat and power through combustion, in (bio)refineries and the pulp and paper industry. Lignin, primarily consisting of phenylpropanoids, can be effectively oxidized to produce aldehydes such as vanillin and syringaldehyde [22,23,24,25]. These aldehydes can undergo condensation reactions with amines to afford imines (also known as Schiff bases) that can be further protonated to afford a range of suitable ILs for various applications. Previous efforts from our lab have been focused on the reductive amination of biomass-derived aldehydes to synthesize renewable ILs that were shown to be very effective for lignocellulosic pretreatment and demonstrated a concept of close-loop biorefinery [26]. Herein, we report, to the best of our knowledge, the first effort to protonate imines formed by direct condensation of lignin-derived vanillin with an amine such as ethylene diamine (EDA). The applicability of these iminium ILs in the processing of lignocellulosic biomass was also explored. Iminium salts are a special class of organic compounds that can be visualized as α-aminocarbocations with an electrophilic nature that might also form a pseudo-base in the presence of water. Such chemical functionality could assist in unfolding several unknown interactions/chemistries when lignocellulosic biomass is considered. It must be highlighted that several examples of various iminium salts exist in the literature for various applications [27,28,29], but the application of renewable iminium ILs in biomass processing remains to be unexplored, until now.

## 2. Results and Discussion

The reaction of an amine with an aldehyde (or ketone) to form an imine via carbinolamine formation was reported by Hugo Schiff, in 1864, as a new series of organic bases [30]. Since then, these imines, also known as Schiff bases, have been investigated in wider contexts including catalysis and bioactive molecules. In the present study, building upon the ease of synthesis, we employed renewable lignin-derived vanillin as an aldehyde precursor, with ethylene diamine (EDA) as an amine source. To obtain vanillin-based Schiff bases, an aqueous solution of vanillin was slowly added to the cold aqueous solution of EDA to dissipate any immediate heat formation. The presence of two amines in EDA offers an opportunity for two unique Schiff bases **1** and **2** simply by the addition of 1 and 2 equivalents of vanillin, respectively, as indicated in Figure 1. The products were obtained in quantitative crude yields as high melting yellow solids (melting points > 230 °C) after filtration and drying in air.

An infrared (IR) spectra of the obtained product evidenced the formation of imine bonds in **1** and **2** as compared with vanillin and EDA (Figure 1). The N-H stretching of the primary amines weakened in **1,** while completely disappearing in **2**, since no primary amine was present in the molecule. Furthermore, the aromatic aldehyde (C=O) stretching @ 1667 cm^−1^ in vanillin exhibited a red shift to 1641 cm^−1^ in **1** and to 1619 cm^−1^ in **2**, a signature for conjugated C=N stretching. The ^1^H and ^13^C NMR analysis of the synthesized Schiff bases in DMSO-*d*_6_ suggested the formation of the desired product (see Materials and Methods).

To prepare the ILs, the obtained Schiff bases **1** and **2** were then treated with acetic acid in 1:1 and 1:2 ratios to protonate the imine N-atom in these bases and yield four unique ILs, i.e., **1A**, **1B**, **2A**, and **2B** (see Figure 1). The synthesis protocol reflects the commonly employed acid–base reaction for the synthesis of a protic IL, where a base (imine in the present case) is mixed with an acid (acetic acid in this study) to expect proton transfer from an acid to a base (degree of proton transfer depends on the physicochemical properties of the reagents) [31,32]. It is important to note that the acetic acid must be added slowly to cold, stirred solution of Schiff bases. The rise in temperature or high concentration of acid (H^+^ ion) results in the hydrolysis of the C=N bonds of Schiff bases to produce water soluble vanillin. The FT-IR spectra of these ILs were recorded to understand the proton transfer (Figure 2). The deprotonation of acid results in a red shift of C=O stretching of carboxylic acid to the C=O stretching of carboxylate. Furthermore, the C-O stretching of carboxylate at 1296 cm^−1^ was observed in all four IL formulations. The peaks around 2600–2700 cm^−1^ have been assigned to strong H-bonding features in the literature [33]. For instance, the intermolecular H-bonding of a concentrated acid has been observed at ~2640 cm^−1^. Such features were also observed for **1** (primary amine) but not for **2** (no primary amine). The partial protonation of amine/imine in **1** resulted in weakening of intensity in **1A,** while the H-bonding seemed to be negligible in **1B**. Similarly, the absence of free amines in **2**, **2A**, and **2B** displayed no signal contributing to strong intermolecular H-bonding in these molecules. We believe, due to the partial protonation of **1**, a dynamic equilibrium exists where the positive charge fluctuates between ammonium (**1A′**) and iminium (**1A**) cations, as shown in Figure 1. The equilibrium is expected to favor protonation of amines over imines as amines are known to be more basic than imines, nevertheless, we do not have any crystallographic or spectroscopic evidence at this point to support the hypothesis.

The differential scanning calorimetry (DSC) profiles of these ILs revealed the preferential proton occupancy sites in 1:1 and 1:2 acetate derivatives of **1** and **2** (Figure 3). The high melting characteristics of imines were also observed in 1:1 acetate derivatives of these imines in the case of **2A**. This indicates that 1:1 salt was not formed, instead, both nitrogen atoms in the imines seemed to be fully protonated in the presence of acetic acid. When one equivalent of acetic acid was added, only 50% of the imine was fully protonated, while the rest remained neutral. This observation was consistent with the FT-IR data where the 1:1 salt displayed characteristic peaks from both imine and 1:2 acetate derivatives (see Figure 2). We hypothesize that no, or full protonation, is not applicable for **1**, since both amine (higher basicity) and imine moieties co-exist in the system. In addition, the thermal gravimetric analysis (TGA) curves of Schiff bases and ILs complement the DSC curves. For instance, the onset of the decomposition of the IL **2B** was observed around 125 °C in both DSC and TGA as compared with the >200 °C for the corresponding Schiff base **2**.

To explore the full chemical potential of lignocellulosic biomass, the fractionation of the strongly held constituent biopolymers (cellulose, hemicellulose, and lignin) in a complex/recalcitrant matrix thorough a pretreatment step is essential. As described earlier, due to the outstanding ability of ILs to dissolve, fractionate, and even convert biopolymers, IL-based pretreatment technologies remain to be attractive for a sustainable biorefinery [26,34,35,36,37]. IL-based pretreatment technologies have been reported to be most effective when operated at temperatures between 90 and 160 °C for 3 h to afford high sugar yields from a given biomass [16,38,39,40,41]. Next, we tested the performance of these ILs for the pretreatment of sorghum biomass at a predetermined condition. For this, 20 wt % of sorghum biomass was mixed with 80 wt % iminium ILs and heated at 140 °C for 3 h. It should be highlighted that all synthesized ILs were solids at room temperature, however, they were expected to melt well below the pretreatment temperature (Figure 3). The pretreatment (PT) efficacy of these ILs are tabulated in Table 1 including solid recovery and sugar release. While considering the amount of dried biomass recovered after pretreatment with ILs followed by washing (see Materials and Methods for details), termed here as solid recovery, all ILs afforded a very high solid recovery in the range of 83–87%. Higher solid recovery at lower pretreatment temperatures are a general trend irrespective of the biomass and ILs employed for pretreatment [36,37,38,39,40,41]. Snapshots of biomass mixed with ILs before and after pretreatment, included in the Table 1, corroborated well with the observed DSC trends, that is, IL **2A** (mixture of **2** and IL **2B**) did not melt completely while all others ILs melted under the pretreatment conditions.

The carbohydrate (glucan and xylan) and lignin amounts of untreated and IL-pretreated sorghum were determined to understand the impact of the pretreatment as a function of the IL. Interestingly, the carbohydrate and lignin contents were found to be similar to that of the untreated biomass. For instance, the glucan, xylan, and lignin contents of the IL-pretreated solids were in the range of 26.1–27.8%, 14.9–15.6%, and 19.1–21.7%, respectively, as compared with the 26.3% glucan, 15.1% xylan, and 19.2% lignin contents in the untreated sorghum biomass. No significant loss of any biopolymer was observed after pretreatment using these ILs, even after considering the solid recovery. Typically, the actual biopolymer removal of carbohydrate and lignin component was calculated as shown below to realize a loss of <10% in all cases:%Removal = [100 − {(%solid recovery) × (composition of the pretreated biomass/composition of the untreated biomass)}](1)

To understand these results, we performed Conductor-like Screening Model for Real Solvent (COSMO-RS) calculations to understand the viability of the IL and biopolymer interactions. In line with previous studies, we calculated the logarithmic activity coefficient (ln(γ)) to predict the dissolution of biopolymers in ILs under investigation [42,43]. We studied the intermolecular interactions between cellulose/lignin and ILs (Figure 4). We did not consider IL **2A** for these calculations as the composition of the synthesized IL was different and the predictions could mislead the experimental observations. In general, lower ln(γ) (i.e., more negative) implies stronger interactions between the solute (cellulose or lignin) and solvent (IL). Based on these predictions, none of the ILs under investigation tended to have a significant interaction with either cellulose or lignin, especially IL **1B**, which had positive ln(γ) implying negligible interactions with both cellulose and lignin. In particular, all these ILs (**1A** and **2B**) had higher affinity for cellulose over lignin. This explains the negligible removal of biopolymers from the biomass after pretreatment with these ILs.

The pretreatment efficacy of these ILs was also measured in terms of carbohydrate digestibility using commercial enzymes, as described in the Materials and Methods section. The enzymatic hydrolysis to release monomeric sugars from the pretreated biomass was carried out at a protein loading of 10 mg per g of biomass, at 50 °C, for 72 h. A high sugar release for ILs **1A**, **1B**, and **2B** was observed affording 69–87% glucose and 63–76% xylose yields (Table 1). As expected, based on the characterization data, low sugar release was observed for IL **2A**. It is worth mentioning that under similar conditions, 38% glucose and 32% xylose yields are achieved after enzymatic saccharification of pretreated solids from 20 wt % sorghum biomass and 80 wt % water mixtures, at 140 °C, for 3 h. 

To further understand the pretreatment mechanism of the Schiff-base ILs, we characterized the pretreated solids from these ILs, especially the solids after pretreatment with IL **2B,** as it afforded the highest sugar release (Table 1). As suggested by the COSMO-RS predictions, these ILs had higher interactions with cellulose (although negligible removal from the biomass); we recorded the powder X-ray diffraction patterns of the untreated and IL **2B** pretreated sorghum to understand the interaction between cellulose and IL (Figure 5, top right). We hypothesized that lower crystallinity of the cellulose might lead to higher enzyme accessibility, however, no such reduced crystallinity was observed for the pretreated solids. The crystallinity of the pretreated solid residue (23.9%) was similar to that of the untreated sorghum (25.5%). Additionally, the TGA of the untreated and pretreated biomass also exhibited similar profiles, other than the removal of extractives including free sugars, aromatics, soluble proteins, among others (expected to be removed during washing of the pretreated biomass) (Figure 5). Typically, most of the previous studies have either considered delignification or reduced cellulose crystallinity (cleavage of intermolecular H-bonds) to explain the pretreatment mechanism [38,39,40]. While some other studies including dilute acid pretreatment have reported efficient sugar release without any significant delignification or reduced crystallinity as observed in the present case [44,45,46]. In addition, enhanced accessible area obtained using Simons’ staining and thermoporosimetry techniques were considered to explain the sugar release after pretreatment [47]. Nevertheless, in the present study, neither of these explains the observed pretreatment efficacy of these ILs.

Finally, we studied the HSQC NMR of the lignin-rich residue obtained after saccharification of pretreated biomass to study the changes in structural features after pretreatment (Figure 6). On comparison with the lignin in the untreated biomass, the observed effectiveness of the Schiff-base ILs for lignocellulosic biomass pretreatment could be examined. For instance, the signals corresponding to the protons on the carbon bearing hydroxyl groups (see A_α_, A_β_, A_γ_, A’_γ_, and B_γ_ structures in Figure 6) either disappeared or weakened after pretreatment with IL **2B** [48,49,50]. This could be due to the chemical interaction of the Schiff-base ILs with the lignin resulting in the abstraction of proton that leads to stripping off lignin from the recalcitrant biopolymer matrix (of cellulose, xylose, and lignin) and rendering active centers on lignin to yield condensed lignin. The aromatic region of the pretreated lignin in the HSQC NMR (Figure 6) supported the formation of condensed lignin [48,49]. These results clearly indicate the chemical interactions of Schiff-base ILs with the lignocellulosic biomass resulting in high sugar yields. We propose that the Schiff-base ILs investigated in this study work mainly by interacting with the lignin component of the lignocellulosic biomass and not cellulose, which remains crystalline after pretreatment. However, the observation of resultant condensed lignin indicates that lignin-carbohydrate linkages in the lignocellulosic biomass were interrupted, leading to increased accessibility to the enzymes, as previously reported for acid pretreatments [45]. To fully exploit the specific application of the Schiff-base ILs for the lignocellulosic biomass utilization for renewable fuels and products, a detailed systematic study is required to gain a better understanding of the mechanism of these ILs in lignocellulosic biomass processing.

## 3. Materials and Methods

### 3.1. Materials

All materials were used as supplied unless otherwise noted. Water was deionized, with specific resistivity of 18 MΩ·cm at 25 °C, from Purelab Flex (ELGA, Woodridge, IL, USA). Choline hydroxide (45% in methanol), acetic acid (>99.7%), sodium hydroxide pellets (≥97%), methanol, sodium azide, and sulfuric acid (98%) were obtained from Sigma-Aldrich (St. Louis, MO, USA). Ethanol (200 proof) was purchased from Decon Labs, Inc. (King of Prussia, PA, USA). Sulfuric acid (72%) was procured from the RICCA chemical company (Arlington, TX, USA). J. T. Baker, Inc. (Phillipsburg, NJ, USA) supplied hydrochloric acid and sodium citrate dihydrate, while citric acid monohydrate (≥99.99%) was obtained from Merck (Kenilworth, NJ, USA). 

Analytical standard grade glucose and xylose were also obtained from Sigma-Aldrich (St. Louis, MO, USA) and used for calibration. 

Sorghum (*Sorghum bicolor*, donated by Idaho National Labs, Idaho Falls, ID, USA) was dried for 24 h in a 40 °C oven. Subsequently, it was knife-milled with a 2 mm screen (Thomas-Wiley Model 4, Swedesboro, NJ, USA). The resulting biomass was then placed in a leak-proof bag, and stored in a dry cool place (4 °C room during the period of use). 

Commercial cellulase (Cellic^®^ CTec3) and hemicellulase (Cellic^®^ HTec3) mixtures were provided by Novozymes, North America (Franklinton, N, USAC).

### 3.2. Syntheses of Schiff Base and Related Ionic Liquids

**Synthesis of Schiff base.** In an oven-dried round-bottomed flask (RBF) containing a Teflon-coated magnetic stirring bar, ethylene diamine was weighed and suspended in water. The flask was mounted on a cold water-bath (5 °C), and an additional funnel was attached to the RBF. An aqueous solution of vanillin was transferred to the addition funnel and added dropwise to the stirred cold aqueous solution of ethylene diamine. The mixture was then stirred for an additional 1 h. The product was obtained after filtration and drying as a yellow solid. Two different ratios of ethylene diamine to vanillin were used to get two different Schiff bases, that is, 1:1 and 1:2. The purity and identity of the synthesized molecules/ILs were determined and established by NMR, IR, and thermal analysis.

Ethylene diamine—Vanillin (1:1), **1**: ^1^H NMR (800 MHz, DMSO-*d*_6_) δ 8.56, 7.30, 7.16, 6.87, 4.01, 3.84, 2.89. ^13^C NMR (201 MHz, DMSO-*d*_6_) δ 162.7, 151.7, 149.9, 133.4, 122.7, 118.2, 113.4, 58.1, 53.1, 40.3.

Ethylene diamine—Vanillin (1:2), **2**: ^1^H NMR (800 MHz, DMSO-*d*_6_) δ 8.61, 7.35, 7.06, 6.84, 4.87, 3.82. ^13^C NMR (201 MHz, DMSO-*d*_6_) δ 162.4, 150.9, 149.8, 132.8, 122.1, 118.7, 113.5, 60.9, 55.4.

**Synthesis of ILs.** In an oven-dried, round-bottomed flask (RBF) containing a Teflon-coated magnetic stirring bar, known amounts of Schiff bases were suspended in water. The flask was mounted on an ice-bath, and an additional funnel was attached to the RBF. Acetic acid (Schiff base/acetic acid, 1:1 and 1:2) was transferred to the addition funnel and added dropwise to the stirred cold suspension of base. The mixture was then stirred for an additional 1 h. The product was obtained after filtration and drying. 

### 3.3. Biomass Pretreatment 

All pretreatment reactions were conducted in duplicate. First, 2 mm sorghum samples and IL were mixed in a 1:4 ratio (*w*/*w*) to afford a biomass loading of 20 wt % in a 15 mL capped glass pressure tube and pretreated for 3 h in an oil-bath heated at 140 °C. After pretreatment, samples were removed from the oil-bath and allowed to cool. Then, 10 mL DI water/ethanol (1:1 *v*/*v*) was slowly added to the biomass-IL slurry and mixed well. The mixture was transferred to 50 mL Falcon tubes and centrifuged at high speed (4000 rpm) to separate solids and remove any residual IL. The ethanol-water washed solid was freeze-dried to obtain dried pretreated biomass for further analysis. 

### 3.4. Enzymatic Saccharification

All enzymatic saccharification was conducted in duplicate. Enzymatic saccharification of pretreated and untreated biomass was carried out using commercially available enzymes, Cellic^®^ CTec3 and HTec3 (9:1 *v*/*v*) from Novozymes, at 50 °C in a rotary incubator (Enviro-Genie, Scientific Industries, Inc., New York, NY, USA). All reactions were performed at 5 wt % biomass loading in a 15 mL centrifuge tube. The pH of the mixture was adjusted to 5 with 50 mM sodium citrate buffer supplemented with 0.02% sodium azide to prevent microbial contamination. The total reaction volume included a total protein content of 10 mg per g biomass. The amount of sugars released was analyzed on an Agilent HPLC 1260 infinity system (Santa Clara, CA, USA) equipped with a Bio-Rad Aminex HPX-87H column and a Refractive Index detector. An aqueous solution of sulfuric acid (4 mM) was used as the eluent (0.6 mL min^−1^, column temperature 60 °C).

### 3.5. Compositional Analysis

All compositional analysis experiments were conducted in duplicate. The compositional analysis of biomass before and after pretreatment was performed using NREL two-step acid hydrolysis protocols (LAP) LAP-002 and LAP-005 [51]. Briefly, 200 mg of biomass and 2 mL of 72% sulfuric acid (H_2_SO_4_) were incubated at 30 °C, while shaking at 200 rpm for 1 h. The solution was diluted to 4% H_2_SO_4_ with 56 mL of DI water and autoclaved at 121 °C for 1 h. The reaction was quenched by cooling down the flasks before removing the solids by filtration using medium-porosity filtering crucibles. The filtrates were spectrophotometrically analyzed for the acid-soluble lignin or ASL (NanoDrop 2000; Thermo Fisher Scientific, Waltham, MA, USA) using the absorbance at 240 nm. Additionally, glucose and xylose concentrations were determined from the filtrate using HPLC (as described previously). The amount of glucan and xylan was calculated from the glucose and xylose content multiplied by the anhydro correction factors of 162/180 and 132/150, respectively. Finally, acid-insoluble lignin was quantified gravimetrically from the solid after heating overnight at 105 °C (the weight of acid-insoluble lignin and ash), and then at 575 °C for at least 6 h (the weight of ash).

### 3.6. Powder X-ray Diffraction

Rigaku MiniFlex 6G 6th Generation Benchtop X-ray Diffractometer equipped with a 600 W sealed source Cu tube and a HyPix-400MF Hybrid Pixel Array 0D/1D/2D detector was used for collecting powder X-ray diffraction (PXRD) data. Data collection and analysis were performed with SmartLab Studio II.

The crystallinity index (CI) was determined from the crystalline and amorphous peak areas of the measured diffraction patterns using the following equation as reported previously [52]:%CI = [(I_002_ − I_am_)/I_002_] × 100(2)
where I_002_ is the intensity of the crystalline plane (002) and I_am_ is the minimum between (002) and (101) peaks and is at about 18°.

### 3.7. COSMO-RS Details

Using the COSMO-RS calculations, the dissolution and/or interaction of cellulose and lignin in the Schiff-base ILs was predicted. To perform these calculations, the initial structures of cellulose, lignin, and ILs were drawn in the Avogadro freeware software [53]. The structures of all the investigated molecules were optimized using the Gaussian09 package at B3LYP (Becke 3-parameter hybrid functional combined with the Lee–Yang–Parr correlation) theory and 6-311+G(d,p) basis set [54,55]. After the geometry optimization step, further, the COSMO file was generated using the BVP86/TZVP/DGA1 level of theory and basis set [56]. The ideal screening charges on the molecular surface were computed using the same level of theory, i.e., BVP86 through the “scrf = COSMORS” keyword [57]. The generated COSMO files were then used as an input in the COSMOtherm (version 19.0.1, COSMOlogic, Leverkusen, Germany) package with BP_TZVP_19 parametrization [58]. In the COSMO-RS calculations, the molar fraction of lignin was set as 0.2, whereas the molar fraction of solvents was set to 0.8 to mimic the experimental pretreatment setup with a biomass to IL loading ratio of 1:4 (*w*/*w*). The activity coefficient of component *i* is associated with the chemical potential *μ*_*i*_ and expressed as [59]:
(3)$$\ln(y_{i})=(\frac{\mu_{i}-\mu_{i}^{0}}{RT})$$
where *μ_i_*^0^ is the chemical potential of the pure component *i*, *R* is the real gas constant, and *T* is the absolute temperature. The details of COSMO-RS calculation are provided in the COSMOtherm’s user manual [60].

### 3.8. FT-IR Analysis

The identities of Schiff base and related ILs was established using FT-IR spectroscopy using a Bruker VERTEX 70/80 system (Billerica, MA, USA). The data were analyzed using Bruker’s OPUS (version 8.2, build 8, 2, 28 (20190310) software.

### 3.9. Thermal Analysis

The thermal behavior was determined using a Mettler Toledo Stare TGA/DSC1 unit (Mettler Toledo, Leicester, UK) under nitrogen (50 mL/min). Samples between 3 and 10 mg were placed in alumina crucibles (70 µL) and heated from room temperature to 800 °C at a heating rate of 10 °C/min to obtain thermal decomposition profiles. Similarly, the Schiff bases and related ILs were sealed in a Hermetic Al pan and the heated from room temperature to 250 °C at a heating rate of 10 °C/min to obtain thermal transition profiles. The data were analyzed using STARe Evaluation software.

### 3.10. HSQC NMR

Untreated and pretreated biomass obtained after enzymatic saccharification were ground with a mixer mill (Qiagen MM300 Mixer, Retsch) using 2 mm diameter stainless steel balls and 30 s^−1^ mixing frequency for 15 min. The ground material was dispersed in DMSO-d_6_ and allowed to stand overnight to extract lignin. The 2D heteronuclear single quantum coherence (HSQC) spectra were collected on a Bruker Avance I 800 MHz spectrometer equipped with a TXI probe at 310K. A standard Bruker pulse sequence (hsqcetgpsisp2.2) was used with the following parameters which are typical for plant cell wall samples. The HSQC spectra were collected from 11 to −1 ppm in F_2_ (^1^H) dimension with 1024 data points for 53 ms acquisition time, and from 165 to −10 ppm in F_1_ (^13^C) dimension with 256 data points for 3.5 ms acquisition time. A total of 256 scans were recorded for each t1 point with a pulse delay of 1 s. The central DMSO solvent peak was used as a reference for the chemical shift calibration for all samples (δ_C_ 39.5 ppm, δ_H_ 2.5 ppm). All HSQC spectra were processed using typical 90° sine square apodization in both F_2_ and F_1_ dimensions and the contours were integrated in the MestreNOVA software (v.14). Peaks were assigned according to published data.

## 4. Conclusions

In summary, we developed lignin-based renewable Schiff-base ILs and explored their application in the lignocellulosic biomass pretreatment. It was noted that imines preferred to be completely protonated, rather than being in a dynamic equilibrium of proton hoping from one iminium center to the other. In addition, the fully protonated iminium IL **2B** was most effective in affording the highest glucose (~87%) and xylose (~76%) release, although negligible interactions with biopolymers were realized based on experimental (no significant removal of biopolymers after water wash) and simulated data. Interestingly, the HSQC NMR spectra suggested changes in the lignin structure after pretreatment with IL **2B,** implying interactions between IL and biopolymers. We would like to emphasize that this work demonstrates a single example of the large number of lignin-derived aldehyde and amine combinations that can be designed and applied for a range of applications including lignocellulosic biomass pretreatment, enabling an overall lower environmental and economic impact. Additionally, rigorous technoeconomic and life cycle models are essential to understand the overall impact and best suited application of the new class of Schiff-base ILs.

## Data Availability

Not applicable.

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
