# Peer review of "Renewable Schiff-Base Ionic Liquids for Lignocellulosic Biomass Pretreatment"

_molecules, 2022, doi:10.3390/molecules27196278_

Round 1
Reviewer 1 Report
1. The discussion of DSC results is poor. The detailed explanation of thermal effects is needed. TGA experiments should be also performed to demonstrate the thermal stability of prepared ILs.
2. Background scattering should be removed from WAXD patterns (Figure 5). Crystallinity of cellulose before and after treatment with synthesized ILs should be calculated, for example, using conventional Segal crystallinity index.
3. Figure 3. The “red” curve should be mentioned in the figure caption. Samples notations (2, 2A etc.) should also be added for more clear reading of manuscript.
4. Some caption of table 1 comes out of the visible border.
Author Response
Please see attachment。

Reviewer 2 Report
Some misspellings can be found.
ALthoug detailed methodology is given, minor issue needs to scpecified: biomass "was stored in a dry cool place". Please specify duration of storage and temperature. Make sure that all the chemicals are mentioned in the 'Materials' section.
Table 1 is not completely visible.
The 'Conclusions' section is very general. It could be useful to include the most important results.
Reviewer 3 Report
In this manuscript, authors have developed lignin-based ILs and explored their application in the lignocellulosic biomass pretreatment. The synthesized ILs were characterized in many ways and the reaction mechanism was revealed. However, three are still some parts in the manuscript to be improved. I recommend this manuscript to be published on Molecules after the following revisions.
1) In the part of Introduction, the peer study on the application of ILs to cellulose treatment has not been fully introduced. It is suggested that the authors consult the literatures related and enrich this part.
2) How to verify the applicability of COSMO-RS model for predicting the activity coefficients of cellulose and lignin in ILs? In my point of view, the authors should collect the data from literature and compare with the predicted results of the model to verify its applicability to such systems. Of course, it is better to verify the model based on the experimental results from this work.
3) The viscosity of the synthesized ILS should be measured, which has an impact on the mass transfer performance.
4) There are impurities in ILs after cellulose treatment. How to remove them for reusing? What is the recycling performance of the ILs synthesized in this work? The author should introduce them in detail.
5) The clearer Figure 6 should be provided. There is a vacancy on the right side of Table 1.
Round 2
Reviewer 1 Report
Authors have revised the manuscript according to comments. I think, the manuscript can be accepted for publication.